# Relation between Body Composition Trajectories from Childhood to Adolescence and Nonalcoholic Fatty Liver Disease Risk

**DOI:** 10.3390/nu16060785

**Published:** 2024-03-09

**Authors:** Gigliola Alberti, Mariana Faune, José L. Santos, Florencia De Barbieri, Cristián García, Ana Pereira, Fernando Becerra, Juan Cristóbal Gana

**Affiliations:** 1Department of Pediatric Gastroenterology and Nutrition, Division of Pediatrics, School of Medicine, Pontificia Universidad Católica de Chile, Santiago 8330077, Chile; gigialberti@gmail.com; 2Department of Nutrition, Diabetes and Metabolism, School of Medicine, Pontificia Universidad Católica de Chile, Santiago 8330077, Chile; mcfaune@uc.cl (M.F.); jsantosm@uc.cl (J.L.S.); 3Radiology Department, School of Medicine, Pontificia Universidad Católica de Chile, Santiago 8330077, Chile; florenciadebarbieri@gmail.com (F.D.B.); cgarcia@med.puc.cl (C.G.); 4Instituto de Nutrición y Tecnología de los Alimentos, INTA, Universidad de Chile, Santiago 7830490, Chile; anitapereiras@gmail.com; 5Independent Researcher, Santiago 8330077, Chile; becerrafernando@gmail.com

**Keywords:** liver steatosis, body composition, fat mass index, fat-free mass Index

## Abstract

NAFLD has become the leading cause of chronic liver disease in children, as a direct consequence of the high prevalence of childhood obesity. This study aimed to characterize body composition trajectories from childhood to adolescence and their association with the risk of developing nonalcoholic fatty liver disease (NAFLD) during adolescence. The participants were part of the ‘Chilean Growth and Obesity Cohort Study’, comprising 784 children who were followed prospectively from age 3 years. Annual assessments of nutritional status and body composition were conducted, with ultrasound screening for NAFLD during adolescence revealing a 9.8% prevalence. Higher waist circumference measures were associated with NAFLD from age 3 years (*p* = 0.03), all skin folds from age 4 years (*p* < 0.01), and DXA body fat measurements from age 12 years (*p* = 0.01). The fat-free mass index was higher in females (*p* = 0.006) but not in males (*p* = 0.211). The second and third tertiles of the fat mass index (FMI) had odds ratios for NAFLD during adolescence of 2.19 (1.48–3.25, 95% CI) and 6.94 (4.79–10.04, 95% CI), respectively. Elevated waist circumference, skin folds, and total body fat were identified as risk factors for future NAFLD development. A higher FMI during childhood was associated with an increased risk of NAFLD during adolescence.

## 1. Introduction

The current obesity epidemic is a major global health problem; the World Health Organization (WHO) considers it one of the most serious public health challenges of the 21st century [1]. As in many countries, in Chile, the number of children who are overweight and obese has steadily increased over time. The prevalence of these conditions in children aged 5–6 years increased from 47.3% in 2011 to 65.8% in 2022 [2]. Nonalcoholic fatty liver disease (NAFLD) is defined as the accumulation of fat exceeding 5% within hepatocytes in the absence of other liver pathologies and chronic alcohol consumption [3], and is considered as the hepatic manifestation of the metabolic syndrome. Epidemiologic data, derived from pediatric studies using noninvasive and invasive tests to diagnose NAFLD, indicate a prevalence of 5% to 10% in the general pediatric population, increasing up to 40% in obese or overweight children [4,5,6]. The highest rate in pediatric patients is in the Hispanic population, from 12% in the general population and up to 52% in adolescents with obesity [4]. Conversely, obesity is distinguished by an excess of adiposity and is intricately linked to the onset of systemic insulin resistance, recognized as a central factor in the pathogenesis of NAFLD. Insulin resistance can trigger an increase in de novo lipogenesis in the liver, leading to a reduction in β-oxidation, ultimately resulting in the accumulation of fat in the liver [7].

Recently, a worldwide consensus panel has recommended replacing the term NAFLD with the term metabolic dysfunction-associated steatotic liver disease (MASLD). MASLD recognizes the intricate link between metabolic dysfunction and liver steatosis, encompassing factors beyond fat accumulation [8].

Childhood is a period of rapid growth with marked changes in body composition [9]. It has been shown that excessive weight gain during childhood is a strong risk factor for the development of NAFLD [10,11]. However, there is scarce information regarding the role of body composition and anthropometric trajectories in the development of this pathology. The main objective of this study was to describe the anthropometry and body composition trajectories throughout childhood and adolescence, in relation to the risk of NAFLD in adolescence.

## 2. Materials and Methods

### 2.1. Study Population

The Growth and Obesity Cohort Study (GOCS) is a longitudinal follow-up initiative that began in 2006, encompassing 1195 children (≈50% female) born between 2002 and 2003. The participants attended 54 public nursery schools in the Southeast area of Santiago, Chile, and are representative of low to middle socioeconomic levels [12]. The study subjects met the specific inclusion criteria, which comprised being a single birth, having a gestational age between 37 and ≤42 weeks, having a birth weight of ≥2500 g, and having no physical or psychological conditions that could significantly impact their growth. The participants were followed annually with assessments that included anthropometric and body composition evaluations. Between 2016 and 2019, participants were recruited prospectively to evaluate the prevalence of NAFLD in adolescence. Participants who presented with the following characteristics were excluded:Previous history of chronic liver disease other than NAFLDSignificant alcohol consumption: approximately 20 g/dayElevation of liver enzymes secondary to drug therapyAny type of malignant disease

### 2.2. Anthropometric and Body Composition Assessment

Weight and height were measured during the annual check-ups, which were conducted from 2006 in the Institute of Nutrition and Food Technology (INTA) by trained dietitians using standardized measurement protocols. The measurements were performed using a digital scale (TANITA 418 BC, precision 0.1 kg; manufactured by TANITA Corporation, Japan, and sourced from IL, USA) and a portable stadiometer (SECA 222, precision 0.1 cm; manufactured by SECA GmbH & Co., Ltd. and sourced from SECA United States). From these data, the body mass index (BMI) was calculated as the ratio of weight (in kg) to height (in m^2^), and the z-score was estimated according to the growth curves of the World Health Organization (WHO) in 2007 [13]. The waist circumference (WC) was measured using a wrap-around metallic tape measure (model W606PM; Lufkin, precision 0.1 cm) just above the iliac crest at the end of a normal expiration. The measurements of the suprailiac, subscapular, biceps, and triceps skinfolds were taken using calipers (Lange caliper, 1 mm graduation). The measurements were performed by grasping the respective fold perpendicular to the index and thumb fingers. The fat mass (FM) and fat-free mass (FFM) were quantified using a bioelectrical impedance analysis using TANITA 418 BC. The FM index (FMI) was calculated as the ratio of the weight of the fat mass (in kg) to height (in m^2^). Likewise, the FFM index (FFMI) was calculated as the ratio of the FFM (in kg) to height (in m^2^). Additionally, body composition was evaluated using DXA (Lunar Prodigy dual-energy X-ray absorptiometry scan). The measurements of weight and height were obtained from the conception of the cohort (approximately 4 years of age) to the time of evaluation. The waist circumference was measured from the age of 4, skinfold measurements were conducted from age 4 to 14 years, bioelectrical impedance analysis was conducted from age 4 year until the time of each evaluation, and DXA scans were performed from age 9 to 13 years in girls and from age 11 to 16 years in boys.

### 2.3. Hattori Charts

Generally, growth is described in terms only of body weight, which is then normalized for height to obtain BMI. This does not consider the deposition of the FM and the FFM and the underlying body compartment changes [14]. Hattori’s body composition charts adjust both FFM and FM for height, which allows the assessment of the nature of weight gain with age in the reference child, and the evaluation of the agreement between BMI and body fatness in samples of subjects of a given age [15]. By correcting the FM and the FFM for height, the nature of the weight gain can be established. Hattori charts were used to describe the trajectory of body composition from age 5 to 15 years, based on the results of the bioelectrical impedance analysis.

### 2.4. Abdominal Ultrasound and NAFLD Diagnosis

NAFLD was defined as an echogenic liver compatible with steatosis in an abdominal ultrasound (US). US was obtained using an Acuson S-2000 unit (6–2 MHz convex and 9–4 MHz linear transducers), where the echogenicity of the liver was compared with the echogenicity of the renal cortex [16]. Two expert pediatric radiologists confirmed the diagnosis. The thickness of superficial and deep intraabdominal fat was also measured using US at the supraumbilical region, according to the previously established method [17,18,19].

### 2.5. Ethics

This research was approved by the Ethics Committee of the School of Medicine of the Pontificia Universidad Católica de Chile (ID:16-030) and of the Institute of Nutrition and Food Technology (INTA) of the Universidad de Chile. Signed informed consent and assent were obtained from the parents and the children, respectively, prior to enrollment.

### 2.6. Statistical Analysis

The association of variables was determined by dividing the participants into two groups: those with a diagnosis of NAFLD (NAFLD group) and those who did not present with the disease (control group). Numerical variables with a normal distribution were expressed as the mean and standard deviation. Variables presenting with an asymmetric distribution with extreme values were shown as median and interquartile range. The Student’s *t* test for independent samples was used to examine the associations of categorical–numerical variables with a normal distribution, and the Wilcoxon rank test was used for those with an asymmetric distribution. The association of categorical variables was evaluated using the chi-square test.

The binary logistic regression model was used to determine the risk of developing NAFLD associated with a higher FMI between 5 and 10 years of age, adjusted for age, sex, maternal pregestational BMI, gestational diabetes (GD), gestational weight gain, and exclusive breastfeeding (EBF) until the sixth month. For this analysis, the numerical variable FMI was transformed into categorical variables expressed in tertiles, with the first tertile serving as the reference.

A significance level of <0.05 was considered for all statistical tests. The statistical power of the study was calculated post hoc because the analyses were performed using the sample size that was initially calculated for the cohort. The FMI and NAFLD analyses obtained a minimum statistical power of 80% with a confidence interval of 95%. Data were analyzed using STATA 15.0 (Stata Corp. 2017. Statistical Software: Release 15. College Station, TX, USA: StataCorp LLC). In addition, the programming language, Pyhton version 3.8, and the IPython libraries were used to run the programs NumPy (version 1.23.2), SciPy (version 1.9.0), and pandas (version 1.4.3) to process and analyze the database and the program, matplotlib, was used to create the charts.

## 3. Results

### 3.1. Nutritional and NAFLD Diagnosis

A total of 784 participants were included (380 were males; average age 15.4 ± 0.98 years; range 13.2 to 17.9 years). The average BMI was 22.3 ± 4.2 for males and 24.4 ± 4.7 for females. In the analyzed sample, 27.5% (216 participants) were classified as individuals who were overweight. Additionally, 12.7% (100 participants) were identified as having obesity, and 2.4% (19 participants) were classified as having severe obesity. The prevalence of NAFLD was 9.8% (77/784). There were no significant differences in the prevalence of NAFLD between males and females (9.2% vs. 10.4%, *p* = 0.577).

At the time of diagnosis, the prevalence of NAFLD was significantly higher in the population of adolescents with obesity (38.1%) compared with adolescents who were overweight (10.3%) and normal weight (2.2%) (*p* < 0.001 for all comparisons). When comparing the characteristics of the participants, the NAFLD group exhibited a higher BMI, higher BMI z-scores, increased waist circumference, and greater amounts of subcutaneous and visceral fat (Table 1).

### 3.2. Anthropometry and Body Composition Trajectories from Childhood to Adolescence

For both males and females, the NAFLD group exhibited elevated BMI z-scores at age 4 years, age 10 years, and age 16 years (*p* < 0.001 for all ages). Additionally, the NAFLD group demonstrated an increased waist circumference from 3 years onward (*p* < 0.05 for all groups), and displayed significantly higher levels of subcutaneous fat across the four skinfold measurements. Significant differences were observed from age 4 to 12 years in both males and females (Table 2).

Regarding the DXA evaluation in males, the NAFLD group had higher levels of total body fat (percentage of fat) and trunk, arm, and leg fat annually from age 12 to 15 years (all *p* < 0.05). Regarding the DXA evaluation females, the NAFLD group exhibited elevated levels of percentage of fat and trunk fat at age 10 years (*p* < 0.05). By the age of 12 years, they demonstrated higher levels of percentage of fat, trunk fat, arm fat, and leg fat (*p* < 0.05).

When we analyzed the trajectory of body composition measured using the bioelectrical impedance analysis using the Hattori charts from age 5 to 15 years, we observed that the NAFLD group had higher FM levels in males and females (*p* = 0.001 and *p* < 0.001, respectively). Regarding the FFM, the NAFLD group had higher values in females (*p* = 0.002), but not in males (*p* = 0.05) (Figure 1a and Figure 2a); the percentage of fat at age 5 years was higher in the NAFLD group in males (*p* = 0.003) and females (*p* < 0.001). Similarly, the NAFLD group had higher FMI values in males (*p* = 0.003) and females (*p* = 0.001). For the FFMI, females with NAFLD had higher values than female controls (*p* = 0.006), but no significant differences were found for males (*p* = 0.206) (Figure 1b and Figure 2b). Comparing the groups by similar weight (Figure 1a and Figure 2a) or similar BMI (Figure 1b and Figure 2b), both groups had different body compositions in terms of all parameters.

### 3.3. Higher Fat Mass Index during Childhood and the Risk of Developing NAFLD in Adolescence

In terms of the FMI, the NAFLD group demonstrated elevated values from age 5 years through adolescence. In males, the disparities were statistically significant at age 5 years (*p* = 0.002), age 10 years (*p* < 0.001), and age 15 years (*p* < 0.001). Likewise, significant differences for females were noted at age 5, 10, and 15 years (all *p* < 0.001). We grouped the sample into FMI increase tertiles during the first 10 years of life, and evaluated the risk of developing NAFLD in adolescence. For the second tertile, the odds ratio (OR) for developing NAFLD in adolescence was 1.92 (1.34–2.74, 95% CI), and for the third tertile, it was 6.12 (4.46–8.39, 95% CI), when compared with the first tertile. After adjusting for age, sex, maternal pregestational BMI, gestational diabetes, weight gain during pregnancy, and exclusive breastfeeding up to the sixth month, the ORs increased to 2.19 (1.48–3.25, 95% CI) and 6.94 (4.79–10.04, 95% CI) for the second and third tertiles, respectively (Table 3).

When we conducted the same analysis based on the annual FMI measurements, the results demonstrated that there is an increased risk throughout childhood, and the risk is highest at the age of 5 years, with an OR of 2.9 (1.77–4.76, 95% CI) (Table 4).

## 4. Discussion

NAFLD has emerged as the primary cause of chronic liver disease in children, and is directly attributable to the substantial prevalence of childhood obesity. This study investigated the associations between body composition trajectories at various stages of infancy and childhood, and the occurrence of NAFLD in adolescence. We found a general prevalence of NAFLD of 9.8%, with no differences according to sex; these findings are similar to other reports in the literature [5,20]. We also found that children who developed NAFLD in adolescence had higher a BMI during childhood, mainly due to a higher FM relative to FFM. A previous study of the same cohort by our group showed that the presence of obesity starting at 2 years of age strongly increased the risk of developing NAFLD in adolescence [11].

Adipose tissue performs several metabolic functions, such as the production of adipokines and cytokines involved in proinflammatory status and extrahepatic injury. Thus, adipocyte hypertrophy may contribute to the development of NAFLD [21,22]. Visceral fat has been reported as an important risk factor for insulin resistance, type 2 diabetes, and cardiovascular disease [23,24]. In contrast, subcutaneous fat of the lower extremities has been associated with an increased sensitivity to insulin [25]. The finding that a higher FM gain during childhood is related to the risk of developing NAFLD in adolescence coincides with the publication of Huang et al. [26], who showed that childhood adiposity trajectories are associated with adolescent insulin resistance, a recognized risk factor for NAFLD.

In this study, subcutaneous fat during childhood, as measured by skinfold testing, was associated with the subsequent development of NAFLD. A cohort of 1167 Australian adolescents showed this association from 3 years of age for the suprailiac skinfold [27]. Similarly, it was found that a larger waist circumference was associated with the development of NAFLD based on data available from age 14 years. In our cohort, we found this association as early as age 3 years. The waist circumference is a simple measurement to obtain and reflects the accumulation of fat in the trunk; however, it does not differentiate between visceral and subcutaneous adipose tissue [28]. The severity of obesity, specifically abdominal obesity, determines a higher risk of NAFLD progression [29]. One cross-sectional study in adolescents with obesity using different body composition measurements showed that waist circumference, trunk fat (measured by DXA), and intra-abdominal fat (measured by ultrasound) predicted the presence of NAFLD [30].

In this analysis, we observed significant differences in body composition at early stages, which emerged as a notable risk factor for NAFLD. Patients diagnosed with NAFLD showed a greater accumulation of fat mass even as early as 5 years of age, and these differences became even more pronounced during adolescence when compared with the control group. These findings suggest that early-life FM accumulation may be associated with an increased susceptibility to NAFLD later in life.

The period of puberty is characterized by various physiological changes, including increased insulin resistance, elevated blood pressure, and changes in cholesterol levels. These factors can contribute to an increased risk of developing metabolic syndrome, which is often associated with the development of NAFLD [31]. Furthermore, we observed marked differences in body composition between males and females in both groups. Females tended to have higher levels of FM, while males exhibited higher levels of FFM. These disparities may be attributed to the differences in sex hormone production [10].

In the evaluation of body composition by DXA, a greater accumulation of body fat and trunk fat during childhood was associated with the subsequent development of NAFLD. Girls who developed NAFLD showed higher levels of body fat, particularly fat centralized in the trunk, at 10 years of age; however, after age 12 years, this accumulation was distributed more homogeneously throughout all compartments. In contrast, boys who developed NAFLD had higher levels of fat, but it was homogeneously distributed throughout all compartments.

When evaluating the highest tertiles of FMI, we found that the accumulation of adiposity from an early age increases the risk of developing NAFLD in adolescence. A study of 2160 adults (34.5% with NAFLD) explored the link between body composition and fatty liver. These findings showed an inverse correlation with fat-free tissue and a direct correlation with fat tissue regarding NAFLD risk. The risk of NAFLD increased when total fat exceeded 32% and 26% in women and men, and abdominal fat surpassed 21% and 13% in women and men, respectively [32]. A study of 100 children with obesity investigated the impact of body composition, particularly the distribution of body fat, and insulin resistance on NAFLD. The results indicated that body fat, particularly abdominal fat, played a role in the development of insulin resistance and subsequent NAFLD [33].

While the assessment of laboratory variables exceeds the scope of the objectives of this study, it is crucial to note that the new definition of MASLD incorporates the determination of cardiometabolic variables, such as plasmatic HDL-cholesterol, plasmatic triglycerides, and fasting serum glucose [8]. Therefore, it is highly relevant to consider these factors in the assessment of metabolic liver dysfunction.

The strengths of this study encompass the prospective gathering of high-quality data, a longitudinal design, an extended follow-up duration, a large participant pool, and the representativeness of the Chilean pediatric population.

One limitation of this study was that the diagnosis of NAFLD relied on ultrasound rather than the gold standard methods of liver biopsy or magnetic resonance imaging with estimated proton density fat fraction. While liver biopsy is considered the most accurate method for detecting NAFLD, it is an invasive procedure that carries risks and is not suitable for large-scale epidemiological studies due to ethical and practical considerations. MRI is a non-invasive technique for assessing liver fat content, but it may be difficult to access in some settings due to its high cost and limited availability. Moreover, both liver biopsy and MRI are not generally considered suitable as screening tests for NAFLD in the pediatric population [34,35]. Another limitation is that the data for the calculation of FMI were only available from age 5 years onwards, preventing the establishment of the initial point of the FMI increase associated with the risk of developing NAFLD in adolescence.

## 5. Conclusions

The trajectories of childhood weight gain and adiposity are associated with the development of NAFLD in adolescence. A larger waist circumference and higher levels of body fat, trunk fat, and subcutaneous fat during childhood are associated with the presence of NAFLD in adolescence. A higher FMI during childhood significantly increases the risk of developing NAFLD in adolescence, with the highest risk at the age of 5 years. Future trials of interventions for controlling adiposity gain during childhood would be helpful in better understanding its effect on NAFLD risk in adolescence.

## Figures and Tables

**Figure 1 nutrients-16-00785-f001:**
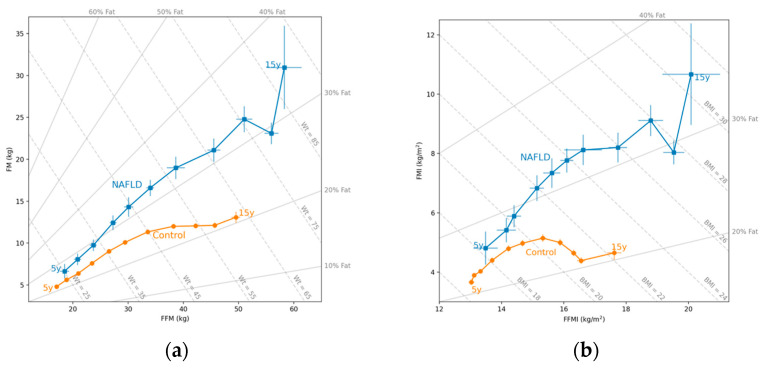
(**a**) Hattori plot for the mean (circles: • squares: ▪) of the fat-free mass (FFM) and the fat mass (FM) in males with NAFLD and in controls. The X axis shows the FFM, and the Y axis shows the FM, both expressed in kg. The diagonal lines indicate the weight (kg) and the percentage of FM (% fat); (**b**) Hattori plot for the mean (circles: • squares: ▪) of the FFM index (FFMI) and the FM index (FMI) in males with NAFLD and in controls. The X axis shows the FFMI, and the Y axis shows the FMI, both expressed in kg/m^2^. The diagonal lines indicate the BMI (kg/m^2^) and the percentage of fat (% fat).

**Figure 2 nutrients-16-00785-f002:**
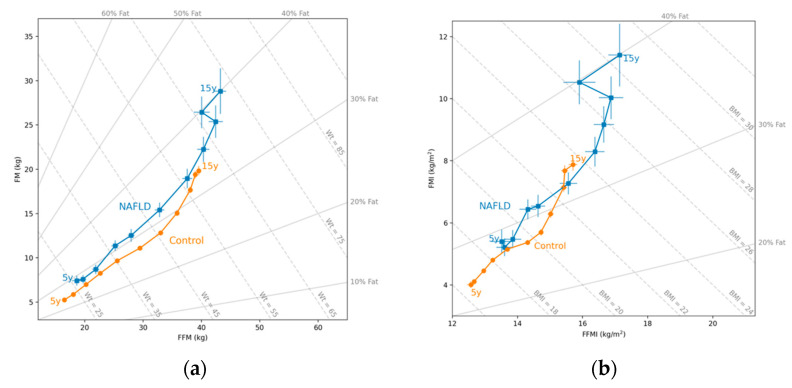
(**a**) Hattori plot for the mean (circles: • squares: ▪) of the fat-free mass (FFM) and the fat mass (FM) in females with NAFLD and in controls. The X axis shows the FFM, and the Y axis shows the FM, both expressed in kg. The diagonal lines indicate the weight (kg) and the percentage of FM (% fat); (**b**) Hattori plot for the mean (circles: • squares: ▪) of the FFM index (FFMI) and the FM index (FMI) in females with NAFLD and in controls. The X axis shows the FFMI, and the Y axis shows the FMI, expressed in kg/m^2^. The diagonal lines indicate the BMI (kg/m^2^) and the percentage of fat (% fat).

**Table 1 nutrients-16-00785-t001:** Anthropometric characteristics of adolescents, by sex.

	Males			Females		
Characteristics	NAFLD Group (*n* = 35)	Control Group (*n* = 345)	*p*-Value	NAFLD Group *(n* = 42)	Control Group (*n* = 362)	*p*-Value
Age (years)	14.9 ± 0.9	15.0 ± 0.94	0.517	15.8 ± 0.9	15.8 ± 0.9	0.985
BMI (kg/m^2^)	28.6 ± 4.7	21.7 ± 3.7	<0.001	30.8 ± 6.7	23.7 ± 3.8	<0.001
BMI z-score	2.2 ± 0.8	0.5 ± 1.1	<0.001	2.2 ± 1.1	0.8 ± 0.9	<0.001
Waist circumference (cm)	92.7 ± 12.1	74.4 ± 9.3	<0.001	88.7 ± 15.3	73.9 ± 8.8	<0.001
Subcutaneous fat (mm)	30.7 (40–22.6)	10.4 (18.7–6)	<0.001	34.2 (49–21.4)	21.6 (29–15)	<0.001
Visceral fat (mm)	45.7 (63.1–39)	37 (45–30.5)	<0.001	45.9 (58.5–34)	34.9 (44–27)	<0.001

The results are presented as *n*, mean and standard deviation or as *n*, median and interquartile range.

**Table 2 nutrients-16-00785-t002:** Waist circumference and skinfolds from childhood to adolescence in groups with and without NAFLD, by sex.

Age (Years)	Measurements	Males		*p*-Value	Females		*p*-Value
		NAFLD Group (*n*)	Control Group (*n*)		NAFLD Group (*n*)	Control Group (*n*)	
6	WC (cm)	(24) 65.3 (9.6)	(242) 58.0 (5.8)	<0.001	(32) 62.5 (7.2)	(251) 57.3 (5.3)	<0.001
	PCSI (mm)	(24) 11.8 (16.2–5.6)	(241) 5.2 (8.7–4)	<0.001	(32) 9.9 (15.5–6.8)	(251) 6.8 (9.9–5.2)	<0.001
	PCSE (mm)	(24) 8.2 (12.6–6.0)	(241) 5.7 (7.3–4.9)	<0.001	(32) 8.8 (12.7–6.8)	(251) 6.5 (8–5.17)	<0.001
	PCB (mm)	(24) 8.3 (9.9–5.6)	(242) 5.0 (6.7–4)	<0.001	(32) 7.5 (9.2- 5.3)	(251) 5.8 (7.3–4.5)	<0.001
	PCT (mm)	(24) 11.2 (15.6–8)	(242) 8.2 (10.7–6.8)	<0.001	(32) 1.5 (14.2–9.4)	(251) 9.7 (11.6–7.7)	0.002
9	WC (cm)	(25) 77.7 (9.4)	(247) 66.6 (8.7)	<0.001	(51) 73.1 (9.4)	(346) 66.1 (8.3)	<0.001
	PCSI (mm)	(25) 24.3(31–18.2)	(247) 13(21.7–7.2)	<0.001	(51) 8.2(36–16)	(346) 16.8 (26.8–10)	<0.001
	PCSE (mm)	(25) 13.3(19.7–10.5)	(247) 7 (11–5.2)	<0.001	(51) 12.7(17.8–8)	(346) 8.7 (12.2–6.2)	<0.001
	PCB (mm)	(25) 11 (15.3–10)	(247) 7 (10.8–4.7)	<0.001	(51) 10.8 (15.5–7)	(346) 7.8 (11–5.8)	<0.001
	PCT (mm)	(25) 18.3(26.2–15.3)	(247) 13 (18–8.8)	<0.001	(51) 18 (22–14.8)	(346) 14 (18.2- 10.5)	<0.001
12	WC (cm)	(35) 88.6 (10.5)	(312) 74.3 (9.8)	<0.001	(37) 84.9 (12.2)	(225) 73.8 (9.3)	<0.001
	PCSI (mm)	(34) 32.5 (42–23)	(304) 16 (24–10)	<0.001	(36) 33 (47- 23.5)	(222) 23 (31–14.3)	<0.001
	PCSE (mm)	(34) 13.5 (17–11)	(304) 8 (11.75–6)	<0.001	(36) 19.8 (25–13)	(222) 12 (17–8.5)	<0.001
	PCB (mm)	(34) 18.5 (25–15)	(304) 11 (16–7)	<0.001	(36) 14.5 (19–11)	(222) 9.8 (13–7)	<0.001
	PCT (mm)	(34) 22.5 (26–16)	(304) 15 (21–10)	<0.001	(36) 24.9 (31–21)	(222) 18.2 (22.7–14)	<0.001

The results are presented as *n*, mean, and standard deviation or as *n*, median, and interquartile range. WC = Waist circumference, PCSI = Suprailiac skinfold, PCSE = Subscapular skinfold, PCB = Biceps skinfold, PCT = Triceps skinfold.

**Table 3 nutrients-16-00785-t003:** Risk of developing NAFLD in adolescence based on FMI tertiles during childhood.

FMI Tertile (kg/m^2^)	OR	95% CI	Adjusted OR *	95% CI
2nd	1.92	[1.34–2.74]	2.19	[1.48–3.25]
3rd	6.12	[4.46–8.39]	6.94	[4.79–10.04]

Note: * Adjusted for age, sex, maternal pregestational BMI, gestational diabetes, weight gain during pregnancy, and exclusive breastfeeding up to the sixth month.

**Table 4 nutrients-16-00785-t004:** Risk of presenting NAFLD in adolescence according to annual FMI during childhood.

Age (Years)	OR	95% CI	Adjusted OR *	95% CI
5–10	1.55	[1.47–1.63]	1.66	[1.55–1.76]
5–5.9	2.90	[1.77–4.76]	2.89	[1.69–4.95]
6–6.9	1.87	[1.54–2.26]	1.80	[1.45–2.22]
7–7.9	1.71	[1.47- 2.00]	1.68	[1.42–2.00]
8–8.9	1.70	[1.50–1.92]	1.73	[1.50–2.00]
9–9.9	1.50	[1.34–1.68]	1.48	[1.31–1.68]
10–10.9	1.55	[1.41–1.71]	1.64	[1.46–1.83]

Note: * Adjusted for age, sex, maternal pregestational BMI, gestational diabetes, weight gain during pregnancy, and exclusive breastfeeding up to the sixth month.

## Data Availability

Part of the data presented in this study are available on request from the corresponding author. The data are not publicly available as they contain sensitive information that requires protection in accordance with privacy regulations and ethical considerations.

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
