# Peer review of "Relation between Body Composition Trajectories from Childhood to Adolescence and Nonalcoholic Fatty Liver Disease Risk"

_nutrients, 2024, doi:10.3390/nu16060785_

Round 1

Reviewer 1 Report

Comments and Suggestions for Authors
  • The authors should consider providing additional details on the research design, including information about sample size determination and any specific methodologies utilized.
  • The manuscript could benefit from clearer explanations of certain technical terms and concepts for readers who may not be familiar with the field.

    Attached is a more specific review report

Author Response

Dear Reviewer,

We appreciate the meticulous comments that have assisted us in enhancing our work. Here, we would like to address, point by point, the issues and concerns raised in response to our manuscript titled "Relation between Body Composition Trajectories from Childhood to Adolescence and Nonalcoholic Fatty Liver Disease Risk."

Kind regards,

Gigliola Alberti 

a) “Longitudinal Data Collection: Ensure consistent and accurate data collection methods over the entire duration of the study to minimize bias and ensure the reliability of the findings”: thanks for your comment. Data collection is carried out in a standardized manner by qualified personnel. Additional information on this point has been added to the methodology.

b) “Control for Confounding Variables: Include additional controls for confounding variables such as diet, physical activity levels, socioeconomic status, and family medical history. Controlling for these factors can help isolate the specific impact of BMI trajectories on the development of NAFLD”: We agree with your comment. Unfortunately, we do not have all the possible comprehensive variables. Regarding socioeconomic status, all participants belong to the same level.

c) “Standardized Measurements: Use standardized protocols and equipment for measuring BMI and diagnosing NAFLD to ensure consistency and comparability of results across different study sites and time points”: The anthropometric measurements and the assessment of fatty liver are protocolized, and standardized equipment is used, all of which is detailed in the study's methodology.

d)”Validation of NAFLD Diagnosis: Consider incorporating liver biopsy or other gold standard methods for diagnosing NAFLD to validate the findings obtained through ultrasound or imaging techniques”: We agree that ultrasound is not the gold standard for diagnosing fatty liver, but liver biopsy and MRI are not generally considered suitable as screening tests for NAFLD in the pediatric population. This is explicitly stated in the study's limitations.

e)“Statistical Analysis: Employ robust statistical methods, such as multivariate regression analysis, to adjust for potential confounders and identify independent associations between BMI trajectories and NAFLD risk”: Thank you for your suggestion. We are presently engaged in developing new statistical analyses that we plan to publish at a later date.

f) “Sample Size Consideration: Ensure an adequate sample size to achieve sufficient statistical power for detecting meaningful associations between BMI trajectories and NAFLD risk, particularly when stratifying the analysis by subgroups or covariates”: The statistical power of the study was calculated post hoc because the analyses were performed using the sample size that was initially calculated for the entire cohort.

g) ”Sensitivity Analysis: Conduct sensitivity analyses to assess the robustness of the findings to different modeling assumptions or exclusion criteria, thereby enhancing the reliability of the results”: Thank you for your suggestion. We are presently engaged in developing new analyses that we plan to publish at a later date.

Reviewer 2 Report

Comments and Suggestions for Authors

The Authors have presented very interesting and detailed analysis of the association between changes in body composition and the risk of NAFLD in children and adolescents. This topic seems to be important, due to the fact that there is an increasing prevalence of overweight and obesity, as well as secondary metabolic disorders in this group. I have only one comment: the Authors did not present any laboratory data of those patients, which is justified by the topic of the paper; however the Authors should describe in the disscusion the new re-definition of NAFLD - MASLD (Metabolic Dysfunction-Associated Steatotic Liver Disease), which was proposed by Delphi consensus in 2023 and according to which cardiometabolic risk factors, including hyperglycemia and dyslipidemia, are also taken into account in the assessment of metabolic liver dysfunction.

Author Response

Dear Reviewer,

We appreciate the meticulous comments that have assisted us in enhancing our work. Here, we would like to address, point by point, the issues and concerns raised in response to our manuscript titled "Relation between Body Composition Trajectories from Childhood to Adolescence and Nonalcoholic Fatty Liver Disease Risk."

Kind regards,

Gigliola Alberti 

a) “The Authors have presented very interesting and detailed analysis of the association between changes in body composition and the risk of NAFLD in children and adolescents. This topic seems to be important, due to the fact that there is an increasing prevalence of overweight and obesity, as well as secondary metabolic disorders in this group”: thank you very much for your comment.

b) The Authors did not present any laboratory data of those patients, which is justified by the topic of the paper; however the Authors should describe in the discussion the new re-definition of NAFLD - MASLD (Metabolic Dysfunction-Associated Steatotic Liver Disease), which was proposed by Delphi consensus in 2023 and according to which cardiometabolic risk factors, including hyperglycemia and dyslipidemia, are also taken into account in the assessment of metabolic liver dysfunction”: We added a new paragraph about MASLD in the introduction and in the discussion.

Reviewer 3 Report

Comments and Suggestions for Authors

Thank you for submitting the manuscript "Relation between Body Composition Trajectories from Childhood to Adolescence and Nonalcoholic Fatty Liver Disease Risk" to Nutrients. Although the work demonstrates a follow-up of a very interesting population sample, the lack of other variables, such as biochemical dosages of liver enzymes or the habit of practicing physical activity, ends up making it difficult to establish a more specific diagnosis. Obviously, this does not take away the merit of the research and the results obtained. Therefore, here are some suggestions to improve the quality of the manuscript:

- Introduction: in general, the introduction needs to be expanded to contain more information necessary to understand the entire text. It needs to be clarified that not every individual with obesity has NAFLD. Another issue is to add the definition of obesity according to the WHO, as in the results and discussion section the authors use this classification for the study individuals.

- Material and methods: it is necessary to check the numerical underwriting throughout the text.

- I think it would be interesting to include in the discussion a survey of 24HR or quality of life of these individuals, that is, to answer the question: why does one part of the sample have obesity/NAFLD and the other not?

Author Response

Dear Reviewer,

We appreciate the meticulous comments that have assisted us in enhancing our work. Here, we would like to address, point by point, the issues and concerns raised in response to our manuscript titled "Relation between Body Composition Trajectories from Childhood to Adolescence and Nonalcoholic Fatty Liver Disease Risk."

Kind regards,

Gigliola Alberti 

  1. “Introduction: in general, the introduction needs to be expanded to contain more information necessary to understand the entire text. It needs to be clarified that not every individual with obesity has NAFLD. Another issue is to add the definition of obesity according to the WHO, as in the results and discussion section the authors use this classification for the study individuals”: We expanded the introduction and added information about NAFLD prevalence. We clarify the WHO definition of obesity in the methodology section.
  2. “Material and methods: it is necessary to check the numerical underwriting throughout the text”: We have reviewed the numerical underwriting.
  3. “I think it would be interesting to include in the discussion a survey of 24HR or quality of life of these individuals, that is, to answer the question: why does one part of the sample have obesity/NAFLD and the other not?”: We agree with your comment, although quality of life and dietary variables are not within the scope of our study objectives.